# Quantification of Microsphere Drug Release by Fluorescence Imaging with the FRET System

**DOI:** 10.3390/pharmaceutics16081019

**Published:** 2024-07-31

**Authors:** Yuying Chen, Huangjie Lu, Qingwei He, Jie Yang, Hong Lu, Jiongming Han, Ying Zhu, Ping Hu

**Affiliations:** 1College of Pharmacy, Jinan University, Guangzhou 511436, China; cyy19981209@stu2021.jnu.edu.cn (Y.C.); tyxx70126@stu2021.jnu.edu.cn (H.L.); heqingwei524@163.com (Q.H.); msjieyang@163.com (J.Y.); luhong0910@foxmail.com (H.L.); 17670960705@163.com (Y.Z.); 2International School, Jinan University, Guangzhou 511436, China; 18203931589@163.com

**Keywords:** drug release, PLGA, microspheres, clozapine, risperidone, FRET

## Abstract

Accurately measuring drug and its release kinetics in both in vitro and in vivo environments is crucial for enhancing therapeutic effectiveness while minimizing potential side effects. Nevertheless, the real-time visualization of drug release from microspheres to monitor potential overdoses remains a challenge. The primary objective of this investigation was to employ fluorescence imaging for the real-time monitoring of drug release from microspheres in vitro, thereby simplifying the laborious analysis associated with the detection of drug release. Two distinct varieties of microspheres were fabricated, each encapsulating different drugs within PLGA polymers. Cy5 was selected as the donor, and Cy7 was selected as the acceptor for visualization and quantification of the facilitated microsphere drug release through the application of the fluorescence resonance energy transfer (FRET) principle. The findings from the in vitro experiments indicate a correlation between the FRET fluorescence alterations and the drug release profiles of the microspheres.

## 1. Introduction

Schizophrenia, an intense psychiatric condition, is distinguished by psychosis, resulting in substantial impairment across personal, familial, social, educational, and occupational spheres. Atypical antipsychotic medications [1], exemplified by risperidone (Ris) and clozapine(Clo), constitute pivotal elements in the management of this disorder [2,3]. Risperidone plays the role of a highly selective antagonism of 5-hydroxytryptamine 2A receptors and dopamine D2 receptors [4], which effectively control the positive symptoms of schizophrenia, such as hallucinations and delusions, while also improving negative symptoms like cognitive impairment and emotional apathy. Adherence to a consistent daily regimen of these pharmacotherapies is essential for symptom management, as discontinuation due to physical factors can lead to reduced efficacy and poor patient compliance [5]. Consequently, there exists a pressing necessity to innovate novel formulations to tackle this challenge.

A diverse array of extended release formulations has been developed to address the requirements of patients for enhanced compliance and sustained dosing [6]. Microsphere formulations are typically formed by active pharmaceutical ingredients (APIs) dispersed as molecules or small clusters in a polymer matrix [7,8]. Poly (lactic-co-glycolic acid) (PLGA) has become extensively utilized in creating microsphere formulations. Its widespread use is due to its highly favorable properties as a carrier for controlled drug release, particularly owing to its excellent biocompatibility and degradability [8,9]. The FDA has approved several PLGA microspheres for slow and controlled release formulations. Risperidone microsphere formulations are available, and there are several generic versions of risperidone microspheres in development [10]. However, the following factors influence the drug release effect of microspheres, including the lactic acid/glycolic acid ratio (LA/GA), molecular weight of PLGA, and variations in the preparation process, all of which can impact the in vivo drug release profile and safety [11,12]. Therefore, further comprehensive investigation into the relationship between drug release kinetics and manufacturing processes is essential to thoroughly investigate the connection between drug release kinetics and manufacturing processes to advance and evaluate both innovative and generic microsphere formulations.

The primary method for monitoring drug release remains the combination of in vitro drug release assays and in vivo blood concentration measurements [13]. However, these testing methods are cumbersome and cannot provide overall data on drug release conveniently. Methods such as scanning electron microscopy (SEM) only provide a static snapshot of the sample and do not capture its dynamic processes [14]. We would like to have a functional tool that can fully reveal and quantify the process of drug release from microspheres. Recently, fluorescent bioimaging has increasingly been used to track nanocarriers, and the routine characterization involves the use of various fluorescent probes. Although common fluorescent probes offer stable and robust signals, they are usually ineffective in providing information on drug release and are optically unstable and prone to photobleaching [15]. Fluorescent dyes with an aggregation-induced burst (ACQ) effect and aggregation-induced luminescence (AIE) properties pose challenges in achieving efficient near-infrared fluorescent probes and have the limitations of fluorescence bursting or reluminescence, making them unsuitable for monitoring the kinetics of drug release from long-acting injectable formulations like microspheres [16,17].

Optical imaging modalities enable the detection of Förster resonance energy transfer (FRET) channels, whose principle is the molecular interaction between donor and acceptor dyes [18]. This method is widely utilized for characterizing the properties of biological and nanomaterials and their responses to biological environments. FRET is highly distance-sensitive, making the encapsulation of both the donor and acceptor within a micrometer carrier essential for achieving high FRET efficiency [19]. The anticipated outcome upon the discharge of the drug from the microspheres into the medium is the diminishment of the FRET signal, with its absence indicating complete release of the drug substance. In addition, when FRET occurs between two fluorescent dyes, a double emission of donor and FRET can be obtained, which opens the possibility of quantitative ratio measurements using fluorescence detection in two different optical windows. To monitor potential drug leakage and microsphere degradation, Cy5 and Cy7 dyes were selected for this study. The dyes were commercially available as cyanine fluorescent dyes, which have advantages such as high penetration in the near-infrared range (700–900 nm). This facilitates increased penetration depth at the excitation wavelength thereby reducing interference of in vivo imaging.

The objective of this study is to establish a platform for quantifying drug release from microspheres in vitro using the Förster resonance energy transfer principle, enabling real-time assessment of drug release [20], a new quantitative technique developed for analyzing drug release from microspheres. As illustrated in Figure 1, we propose to use the FRET effect relationship between Cy5 and Cy7 to realize drug release quantification when risperidone or clozapine drugs are released from microspheres. The results show that the FRET principle successfully enables the visualization and quantitative monitoring of microspheres. The fit of the in vitro release pattern and the FRET alteration pattern showed superior congruence for Clo-FRET-PLGA/M compared to Ris-FRET-PLGA/M. These results highlight the utility of the FRET principle in therapeutic diagnostic drug delivery with the potential in using fluorescence readout for quantitative monitoring of the drug release process.

## 2. Materials and Methods

### 2.1. Materials

Poly (D,L-lactic acid-glycolic acid copolymer) (PLGA) copolymer (molecular weight: 30k Da) was purchased from Jinan Daigang Bioengineering Co., Ltd. (Jinan, China) with a PLA/PGA ratio of (*w*/*w*) 75:25. Poly(vinyl alcohol) (87–89% hydrolyzed) was purchased from Acmec Biochemistry Co., Ltd. (Shanghai, China); Cy5 was purchased from Shanghai Yuanye Bio-technology Co., Ltd. (Shanghai, China); Cy7 was obtained from Nanjing Guye Ltd. (Nanjing, China). Trifluoroacetic acid (TFA), formic acid (HCOOH), acetonitrile (CH_3_CN), and the drugs (i.e., risperidone and clozapine) were purchased from Shanghai McLean Biochemical Co. (Shanghai, China). Dichloromethane and dimethyl sulfoxide (DCM, DMSO) were purchased in analytical grade Shanghai Anergy Co., Ltd. (Shanghai, China). ddH_2_O was used for any required steps.

### 2.2. Preparations of Clo-FRET-PLGA/M and Ris-FRET-PLGA/M

The emulsification solvent volatilization method was utilized for the preparation of microspheres. PLGA with a molecular weight of 30k Da was specifically selected for this purpose, with a PLGA to model drug (risperidone and clozapine) ratio of 5:1. The ratio of PLGA to fluorescent dye was 1000:1. Fluorescent dyes, pharmaceuticals, and PLGA were dissolved in dichloromethane to form the organic phase, and a volume ratio of 1:14 was maintained between the oil phase and the aqueous phase. Utilizing a homogenizer operating at a speed of 3400 rpm/min for 2 min, the organic phase was gradually introduced into a specified volume of 1% PVA in the aqueous phase, yielding an O/W primary emulsion under the controlled delivery of the organic phase by a syringe pump.

The emulsion underwent transfer into an aqueous PVA solution of 0.5% (*w*/*v*), at a volume three times that of the emulsion, with stirring at 1000 rpm [21]. A vacuum was subsequently applied to the aqueous phase for 4 h at 1000 rpm to facilitate the evaporation of methylene chloride and the solidification of the microspheres. Following this, the resulting suspension was transferred to a 50 mL centrifuge tube and centrifuged at 4000 rpm for 5 min. The microspheres were then subjected to three washes with 20 mL of deionized water and dried using a freeze dryer [14]. The resultant white powder, signifying the completion of the microsphere preparation process, was stored at 4 °C until further use [22].

### 2.3. Determination of Drug Loading and Encapsulation Efficiency of Clo-FRET-PLGA/M and Ris-FRET-PLGA/M

For the formulation of drug-loaded microspheres, approximately 5 mg of both Clo-FRET-PLGA/M and Ris-FRET-PLGA/M was accurately measured and dissolved in 2.5 mL of dimethyl sulfoxide (DMSO). Following the complete disruption of the microspheres, the solution was moved to a 10 mL volumetric flask and adjusted to its final volume using methanol. The mixture was subjected to ultrasonic agitation in an ultrasonic cleaner until dissolution was confirmed. Filtration was performed using a Millex^®^ HV, 0.22 μm polyvinylidene fluoride (PVDF) syringe filter to ensure clarity [23].

The drugs encapsulated within the microspheres were quantified through high-performance liquid chromatography (HPLC). Employing an Agilent 1260 HPLC system, coupled with a SuperLu C18-AQ column (250 × 4.6 mm, 5 μm, 100 Å), facilitated the establishment of a drug concentration profile for risperidone and clozapine. The concentration of risperidone in Ris-FRET-PLGA/M was ascertained with a mobile phase of acetonitrile and water/trifluoroacetic acid (TFA) in a ratio of 30/70/0.1 (*v*/*v*/*v*), at a flow rate of 1 mL/min and a detection wavelength set at 275 nm. Similarly, the concentration of clozapine in Clo-FRET-PLGA/M was determined using a mobile phase of methanol and water/TFA in a ratio of 65/35/0.5 (*v*/*v*/*v*), with a consistent flow rate of 1 mL/min and a detection wavelength set at 254 nm [24].

Subsequent to the HPLC analysis, the concentrations of risperidone and clozapine were deduced from the established standard curves. These data enabled the calculation of drug loading (DL) and encapsulation efficiency (EE) for the microspheres. To ensure accuracy and reproducibility, each formulation underwent triplicate analysis, with the outcomes expressed as mean values accompanied by standard deviations.
(1)$$\mathrm{Encapsulation}\;\mathrm{efficiency}\;(\%)=\;\frac{\mathrm{actual}\;\mathrm{amount}\;\mathrm{of}\;\mathrm{encapsulated}\;\mathrm{drug}\;\mathrm{of}\;\mathrm{microspheres}}{\mathrm{theoretical}\;\mathrm{amount}\;\mathrm{of}\;\mathrm{encapsulated}\;\mathrm{drug}\;\mathrm{of}\;\mathrm{microspheres}}\times 100$$
(2)$$\mathrm{Drug}\;\mathrm{loading}\;(\%)=\frac{\mathrm{weight}\;\mathrm{of}\;\mathrm{drug}\;\mathrm{contained}\;\mathrm{in}\;\mathrm{microspheres}}{\mathrm{weight}\;\mathrm{of}\;\mathrm{microspheres}}\;\times 100$$

### 2.4. In Vitro Drug Release

To ascertain the release of risperidone or clozapine from the microspheres, 5 mg of Ris-FRET-PLGA/M and Clo-FRET-PLGA/M was weighed and dispersed in 4 mL of phosphate-buffered saline (PBS, pH 7.4) containing 0.05% NaN_3_ as preservative and 0.02% (w/v) Tween 80. The release experiment was carried out under sink conditions, and 0.02% Tween 80 served to increase the wettability and prevent the microspheres from floating. The microspheres were placed in a thermostatic oscillator (ZD-85, manufactured by Xuri Experimental Instrument Factory Co., Ltd., Changzhou, China), and the microspheres were kept at 37 °C and a shaking rate of 100 rpm. On the initial day 0, day 1, and every two days, 3 mL of samples was taken in 4 mL Eppendorf microtubes and centrifuged at 4000 rpm for 5 min at predetermined time points. To determine the drug content of the samples, 3 mL of supernatant was withdrawn and replaced with 3 mL of fresh release medium in the released samples. A plot of cumulative percent release versus time, calculated using the following equation, allowed us to assess the cumulative drug release of risperidone or clozapine from the microspheres [25]. Each formulation was also analyzed in triplicate, and the mean and standard deviation were reported. The calculation formula was (3):(3)$$\mathrm{Release}\;\mathrm{rate}\;\mathrm{of}\;\mathrm{microspheres}\left(\%\right)=\frac{\mathrm{cumulative}\;\mathrm{release}\;\mathrm{of}\;\mathrm{drug}\;\mathrm{from}\;\mathrm{microspheres}}{\mathrm{total}\;\mathrm{content}\;\mathrm{of}\;\mathrm{drug}\;\mathrm{in}\;\mathrm{microspheres}}\;\times 100$$

### 2.5. Detection of Fluorescence Intensity

During the in vitro release experiment described in Section 2.4., the fluorescence intensity was measured by aspirating the whole solution and microspheres on the initial day 0, day 1, and every two days thereafter until the fluorescence intensity disappeared completely.

A fluorescence spectrometer (FS5, Edinburgh Instruments, Livingston, UK) was utilized to measure the fluorescence intensity under the excitation wavelength of 640 nm, and the absolute values of emitted fluorescence intensity at 690 nm and 790 nm were subsequently recorded. The decreasing trend of FRET fluorescence for each measurement was calculated using F/D using day 0 of the initial phase as a 100% reference.

The calculation formula was (4) [26].
(4)$$F/D=\frac{F_{FRET}}{F_{Donor}}$$

F_FRET_: the FRET fluorescence intensity, (Ex = 640 nm, Em = 790 nm); F_Donor_: the donor fluorescence intensity, (Ex = 640 nm, Em = 690 nm).

Equations (5) and (6) were used as a 100% reference using day 0 of the initial phase and each subsequent time after ratio processing from the absolute value of the measured fluorescence intensity:(5)$$\mathrm{FRET}\;\mathrm{change}\;\mathrm{rate}\;(\%)=\frac{F/D}{{F0/D}_{0}}\times 100\%$$
(6)$$\mathrm{FRET}\;\mathrm{remaining}\;\mathrm{rate}\;(\%)=(1\;-\;\frac{F/D}{{F0/D}_{0}})\;\times 100\%\;$$

### 2.6. Determination of Weight Loss of PLGA Microspheres

The weight of the microspheres (W_0_) was recorded before putting them in 4 mL of release medium for the release of Ris-FRET-PLGA/M on days 1, 4, 7, 14, 18, and 26. Similarly, after placing Clo-FRET-PLGA/M in the release medium for 1, 4, 7, 14, 22, and 30 days, the microspheres and medium were vacuum-filtered through a 0.8 μm microporous filter membrane rinsed with a small quantity of purified water. The resulting microspheres underwent drying in a vacuum drying oven at 30 °C for a duration of 24 h until they were completely dry. The weight of the dried microspheres was recorded as W_2_ [27]. The weight loss of the microspheres was calculated as follows (7):(7)$$\mathrm{Mass}\;\mathrm{Loss}\;\left(\%\right)=\frac{\left(W_{0}\;-\;W_{2}\right)}{W_{0}}\times 100\%$$

### 2.7. Determination of Water Absorption of PLGA Microspheres

Ris-FRET-PLGA/M was released in the release medium on days 1, 4, 7, 14, 18, and 26. Similarly, after days 1, 4, 7, 14, 22, and 30 of Clo-FRET-PLGA/M exposure to the release medium, the microspheres and the release medium were filtered with a 0.8 μm microporous filter membrane under vacuum filtration. Subsequently, a minimal quantity of distilled water was applied, saturating the microspheres, which were then subjected to vacuum drying for 1 min to eliminate surface moisture, and the moisture content of the microspheres was weighed immediately This weight, representing the water-containing microspheres, was recorded as W_1_. The resulting microspheres underwent drying in a vacuum drying oven at 30 °C for a duration of 24 h until the weight of the microspheres was constant and not decreasing, and the weight of the dried microspheres was weighed and recorded as W_2_. Then, the difference in the weights between the wet microspheres and the dried microspheres was used for the estimation of the interparticle water ratio (W_i_) at time t, which was defined as (8):(8)$$W_{i(t)}=\frac{\left(W_{1(t)}\;-\;W_{2(t)}\right)}{W_{2}(t)}$$

### 2.8. Confocal Laser Microscopy Scanning

The release medium conditions for both microspheres were the same as in Section 2.4, with three samples taken from each of the seven time points, 1 d, 4 d, 7 d, 14 d, 18 d, and 26 d for Ris-FRET-PLGA/M. Similarly, Clo-FRET-PLGA/M took three samples out at each of the six time points of 1 d, 4 d, 7 d, 14 d, 22 d, and 30 d. After the samples were filtered through a 0.8 μm microporous membrane filter and subsequently rinsed with a modest quantity of purified water, the resulting microspheres were placed in a vacuum drying oven (DZF-6050AB, Shanghai Li-Chen Bangxi Instrument Technology Co., Ltd., Shanghai, China) for drying, and the fluorescence changes in the microspheres were obtained as a graph of changes in fluorescence over time.

The collected microspheres were imaged using an ultrafast laser confocal microscope (FV3000, Olympus Corporation, Tokyo, Japan) [28]. The conditions of the instrument were set as follows: 1. Donor channel: Ex = 640 nm, Em = 690 nm. 2. FRET channel: Ex = 640 nm, Em = 790 nm.

### 2.9. Scanning Electron Microscope (SEM)

In order to investigate the changes in the morphological characteristics of the two microspheres over time, three samples of Ris-FRET-PLGA/M were taken out at each of the seven time points, 1 d, 4 d, 7 d, 14 d, 18 d, and 26 d, after incubation in PBS at 37 °C. Similarly, three samples of Clo-FRET-PLGA/M were taken out at each of the six time points of 1 d, 4 d, 7 d, 14 d, 22 d, and 30 d for SEM evaluation. Conductive double-sided adhesive tape was adhered to a standard sample carrier stage. Following this, a limited quantity of microspheres earmarked for testing were positioned on the conductive tape. Any unattached microspheres were expelled using a high-speed airflow. The microsphere samples were then sputter-coated with gold before the carrier stage was introduced into a high-resolution scanning electron microscope (SU1000, Hitachi, Tokyo, Japan) for observation of the microspheres’ surface morphology [29].

### 2.10. Relation Study

The risperidone or clozapine release curves measured in vitro were compared with their corresponding (1-F/D) curves; the f_2_ similarity factor was calculated; the correlation between the two curves from the same microspheres was established by using Origin software (Version 9.1, OriginLab Corporation, Northampton, MA, USA) and fitted equations were obtained. The drug release from microspheres was predicted by determining f_2_ for the similarity coefficient when (f_2_ > 50 is considered as valid value) [30].

## 3. Results

### 3.1. Preparations and Characterizations of Ris-FRET-PLGA/M and Clo-FRET-PLGA/M

The experimentally determined drug loadings of Ris-FRET-PLGA/M and Clo-FRET-PLGA/M in this study were 18.52 ± 1.26% (*w*/*w*) and 16.40 ± 1.21% (*w*/*w*), respectively. The encapsulation rates were all similar, approximately 73% for both samples. SEM images of Ris-FRET-PLGA/M and Clo-FRET-PLGA/M are shown in Figure 1. The microspheres exhibited spherical shapes with uniformly smooth surfaces across all samples. Their particle sizes were measured and are shown at around 70 μm.

Fluorescent dyes Cy5 and Cy7 were selected as FRET pairs for encapsulation into microspheres. Their chemical structures and physical properties are depicted in Appendix A. These dyes were chosen due to their similar log *p* values with the model drugs and their ideal in vivo imaging spectral properties. The fluorescence spectra of the fluorescent dyes under methanol solution were recorded as in Figure 2A, with strong emission bands at 650–750 nm for Cy5. As shown in Figure 2B, Cy7 has a strong emission band at 750–850 nm. As depicted in Figure 2C, the absorption spectrum of Cy5 aligns precisely with the emission spectrum of Cy7, while its emission in the near-infrared region at 780 nm is completely separated from that of Cy5; thus, theoretically, the energy of Cy5 can be efficiently transferred to the Cy7 fluorescent dye with the generation of the FRET effect, which makes this pair of combinations perfectly suited for ratiometric FRET imaging. As illustrated in Figure 2D, Cy5 and Cy7 were encapsulated into microspheres resulting in emission peaks corresponding to the donor channel, as well as remarkably intense FRET peaks. The results confirm that the microspheres exhibiting the FRET effect were successfully prepared.

### 3.2. In Vitro Drug Release

The in vitro release profiles of both microspheres, conducted under simulated physiological conditions mimicking in vivo environments, demonstrated an initial burst release (within 24 h) of risperidone and clozapine, which accounted for less than 5% of the total release [31]. Subsequently, sustained drug release was observed, as illustrated in Figure 3A. The release rate of risperidone exhibited slow–fast–slow phases, finally reaching a plateau of 94% release by day 32. Figure 3B illustrates the donor and FRET fluorescence intensities obtained from Ris-FRET-PLGA/M, as measured using a fluorescence photometer. On day 2, both donor and FRET fluorescence intensities began to increase, followed by a continuous decline after day 6. This phenomenon might be due to the fact that the concentrations of donors and acceptors were relatively high within the microspheres, which causes quenching by the aggregation of fluorescent dyes, followed by a decrease in fluorescence concentration within the microspheres due to the beginning of the release of the drug and fluorescent dyes from the microspheres, and the quenching effect is diminished. The microspheres show a reluminescence phenomenon, and the fluorescence intensity value increases. The phenomenon suggests that the concentrations of fluorescent dyes could be further optimized to obtain a more ideal FRET change profile in future studies.

Despite the decreasing trend observed in the later stages, the absolute fluorescence intensities of the dyes fluctuated due to instrumental and other interferences. Therefore, through the computation of the FRET fluorescence intensity ratio to that of the donor fluorescence intensity (F/D), depicted in Figure 3C, the fluctuations in FRET of risperidone microspheres remained minimal, illustrating a consistent downward trend. Therefore, compared to the absolute fluorescence intensities of the donor and FRET channels, the FRET changes in the microspheres were more accurately characterized after applying the F/D ratio. Figure 3D indicates that clozapine’s release was obviously faster, reaching 50% by day 6, after which the release rate significantly slowed down until complete release on day 26. Figure 3E presents the donor and FRET fluorescence intensities measured for Clo-FRET-PLGA/M. On day 2, both donor and FRET fluorescence intensities began to increase, followed by a steady decline starting on day 4. This pattern is analogous to the FRET change results observed with risperidone, although the fluorescent intensities for Clo-FRET-PLGA/M declined faster compared to Ris-FRET-PLGA/M. Figure 3F shows that after F/D calculation, a more stable decreasing trend in FRET change for clozapine microspheres was obtained [32].

Figure 4A,B depict the comparison of fluorescence changes and drug release for Ris-FRET-PLGA/M and Clo-FRET-PLGA/M, revealing that the similarity for clozapine is superior to that of risperidone microspheres. This similarity was calculated and is shown as the f_2_ value, which was 59.79 for clozapine (Table 1), confirming that the FRET change in clozapine has a better fit with its drug release compared to risperidone.

While risperidone had a lower log *p* value than clozapine, risperidone exhibited a slower release profile compared to clozapine. The degradation mechanism of PLGA involves chemical hydrolysis and enzyme-catalyzed hydrolysis [33]. The pH value of the environment greatly influences the hydrolysis and degradation of PLGA molecules. The water solution of risperidone or clozapine appears to be alkaline due to their chemical structures, and the hydrolysis of PLGA is accelerated in an alkaline environment. As in Figure 4C, the protonated amine groups on drugs can induce alkaline catalysis and accelerate the degradation of PLGA. More precisely, the tertiary amine molar concentration of risperidone is calculated to be 0.00277 g/mol, while that of clozapine is 0.00573 g/mol, which suggests that clozapine is a more effective catalyst for PLGA polymer degradation. Finally, due to its molecular weight being around 100 Da smaller than risperidone, clozapine molecules could diffuse more easily through the microspheres’ networked pores, which lead to a faster release profile.

### 3.3. Determination of Weight Loss and Water Absorption of PLGA Microspheres

Microspheres require water absorption to initiate hydrolytic degradation. The expedited release observed in the release phase under specific conditions is believed to be associated with the ingress of water into the hydropore network of the microspheres. As depicted in Figure 5A,B, notable disparities were observed in the water uptake profiles of Ris-FRET-PLGA/M and Clo-FRET-PLGA/M. They both showed a gradual increase in water absorption over time, although clozapine absorbs water more rapidly. This trend is similar to the in vitro drug release results.

The dissolution of polymers represents a crucial mechanism influencing the release of drugs from PLGA microspheres. Assuming erosion as the primary release mechanism and the drug is uniformly dispersed within the microspheres, the rate of drug release would be similar to the rate of polymer erosion (i.e., the rate of polymer mass reduction) [21,34]. Data depicting the mass loss during the in vitro release for Ris-FRET-PLGA/M and Clo-FRET-PLGA/M are presented in Figure 5C,D. The Ris-FRET-PLGA/M dissolved significantly slower than Clo-FRET-PLGA/M in the in vitro environment. When risperidone was completely released, the mass loss of microspheres reached 45%. In contrast, the mass loss of Clo-FRET-PLGA/M continued to increase with time and finally reached 90%.

### 3.4. Degradation of Ris-FRET-PLGA/M and Clo-FRET-PLGA/M over Time by Scanning Electron Microscope (SEM)

The morphological changes in degraded Ris-FRET-PLGA/M and Clo-FRET-PLGA/M are illustrated in Figure 6. At day 0 and day 1, both microspheres maintained smooth surfaces and exhibited spherical shapes. Subsequently, the Clo-FRET-PLGA/M displayed earlier morphological changes, as some of the microspheres showed initial degradation. This process is due to the hydrolysis of PLGA molecules and is expedited by the presence of the amine groups on clozapine, wherein PLGA hydrolyzed to lactic acid and glycolic acid under humid conditions [35]. In contrast, risperidone microspheres only exhibited cracks and pores on day 7 with a wrinkled surface, while maintaining their intact spherical shape. Over time, the risperidone microspheres showed an increase in pore size by day 14, and the particle size seemed to increase due to water absorption and swelling, accompanied by the appearance of some fragments. By day 30, the microspheres demonstrated increased surface porosity, consistent with the in vitro drug release results.

### 3.5. Degradation of Ris-FRET-PLGA/M and Clo-FRET-PLGA/M over Time by Confocal Laser Microscopy Scanning

Subsequently, confocal imaging was employed to observe the changes in the FRET effect with microsphere degradation, as depicted in Figure 7. Both donor and FRET channels were observed and recorded. The results demonstrate the presence of the FRET effect of the two types of microspheres. The fluorescence intensity of donor channels and FRET channels of both microspheres decreased with time. By the seventh day, some microspheres displayed cavities and uneven fluorescence distributions. Furthermore, a gradual decrease in fluorescence and loss of spherical shape were noted, especially for clozapine microspheres, which degraded at a faster rate than risperidone. Ratiometrically processed images indicate that the final FRET/D signal change closely mirrored the in vitro drug release, which was related to the PLGA degradation and microsphere fragmentation. These observations confirm the visualization and quantification of changes in FRET fluorescence, correlating with drug release from the microspheres.

### 3.6. Correlation and Prediction Studies

A series of in vitro experiments demonstrated the relationship between FRET changes in microspheres and in vitro drug release. By fitting the FRET changes to drug release data, we developed equations that provided a better quantitative understanding of this relationship. Figure 8A,C show a one-to-one linear relationship between the in vitro drug release and FRET changes for both types of microspheres. The correlation for Clo-FRET-PLGA/M was more significant with an R2 value of 0.983. Using the correlation formulas derived from our analysis (Table 2), FRET change data can be used to predict the drug release profiles in vitro. Figure 8B,D depict the actual and predicted in vitro drug release profiles of both microspheres. The trends of these predicted release data closely align with their observed data. Our analysis of the f_2_ values indicates that a more accurate in vivo drug release profile could be obtained using the FRET change data and the correlation equation, particularly for Clo-FRET-PLGA/M, which showed a stronger correlation. Furthermore, the findings also lend quantitative backing to the theory that FRET alterations can indicate in vitro drug release from microspheres.

## 4. Conclusions

In conclusion, this study represents a novel approach to quantify released drug from microspheres based on FRET principle. Our research successfully shows the preparation of PLGA microspheres encapsulating two fluorescent dyes, Cy5 and Cy7, along with two model drugs (risperidone and clozapine) to leverage the FRET concept in imaging microsphere drug release applications. The application utilized the fluorescence ratio of FRET and donor to characterize the FRET fluorescence changes within the microspheres, enabling accurate, sensitive, and reproducible quantitative analysis. Successful mitigation of interferences in common fluorescence imaging was achieved, and a reliable calibration of the results was obtained. Furthermore, this study compared the results of different drug release profiles, revealing that clozapine microspheres and the selected FRET pairs exhibited superior correlation, whose in vitro correlation between fluorescence changes and drug release reached 0.98. This work significantly contributes to the assessment and advancement of the drug release from PLGA microspheres, and expands avenues for further investigation into the mechanisms of PLGA-based long-acting products.

## Data Availability

The data presented in this study are available in this article and Appendix A.

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
