# Peer review of "Quantification of Microsphere Drug Release by Fluorescence Imaging with the FRET System"

_pharmaceutics, 2024, doi:10.3390/pharmaceutics16081019_

Round 1

Reviewer 1 Report

Comments and Suggestions for Authors

The manuscript submitted by Chen et al. reports on the qualitative and quantitative analysis of drug encapsulation/release performance of PLGA microparticles. The Authors present their methodology based on fluorescence resonance energy transfer (FRET) principle to determine the drug release kinetics.

I would like to emphesize that the amount of data and the variety of experimental methods presented in this manuscript are impressive. Yet, the main hypotheses presented in this manuscript, which is based on the FRET method, has serious flaws. FRET is a very sophisticated physiochemical mechanism that can indeed provide important insides in biomedical studies. However, what is presented by the Authors in this manuscript is quite an unusual approach to the use of FRET method. In a conventional approach where the FRET is employed as a method to analyze the drug release kinetics, one of the drugs should play the role of donor or acceptor to determine the translocation of the drugs. However, the way that the Authors employed FRET method does not really provide any translocation information about the drug. It could be probably called as a "passive" or "indirect" FRET method and very unusual approach to use the FRET function.

I have the following questions to the Authors to clarify their methodology in this manuscript, which should be answered before the manuscript can be considered for a publication:

1. As far as I understood, the drugs were packed into the PLGA spheres along with the FRET donor/acceptor drugs. How did Authors make sure that the drugs were released from the spheres a long with the FRET pair, in the same rate with the FRET pair? How does the FRET signal translates into the drug release kinetics? If the drugs themselves do not play any role in generation of FRET signal, what is the actual role of FRET dye pairs in determining the drug release rate?

2. There are many other methods that could be used to determine the amount of drug released from the microspheres. For instance, a simple HPLC analysis of the dialysates of the PLGA spheres could work perfectly. What are the advantages of using FRET methodology (and a qualitative/quantitative analysis using FRET function) over any other conventional approach?

3. Looking at the way that the Authors employed the FRET method, it seems to me that the FRET analysis provides more information on the disassociation of the PLGA sphere rather than the release kinetics of the drugs. Should Authors reconsider what is actually quantified here? 

Comments on the Quality of English Language

The language must be grammatically improved and some certain parts of the text must be restructured using more punctuations.

Author Response

  1. As far as I understood, the drugs were packed into the PLGA spheres along with the FRET donor/acceptor drugs. How did Authors make sure that the drugs were released from the spheres a long with the FRET pair, in the same rate with the FRET pair? How does the FRET signal translates into the drug release kinetics? If the drugs themselves do not play any role in generation of FRET signal, what is the actual role of FRET dye pairs in determining the drug release rate?

Author Response: Thank you for your questions. Actually, the aim of this study is to explore the possibility to use the FRET changes to represent the drug release. We only discovered that the drug release of the two drugs could be approximately represented by the FRET changes, although not at the same rate. Clozapine showed a more similar trend of drug release to the FRET changes compared to risperidone. We concluded from the results that the FRET effect was heavily influenced by the erosion of the PLGA matrix, as well as the degradation of the microsphere. As the drug release from the PLGA microspheres was dominated by erosion mechanism, therefore, the FRET changes of the dyes could be used to indirectly determine the drug release rate.

  1. There are many other methods that could be used to determine the amount of drug released from the microspheres. For instance, a simple HPLC analysis of the dialysates of the PLGA spheres could work perfectly. What are the advantages of using FRET methodology (and a qualitative/quantitative analysis using FRET function) over any other conventional approach?

Author Response:Currently, conventional in vitro and in vivo methods are cumbersome and complex, and cannot detect drug release in real time. Using the FRET method in this study to quantify drug release, it can visualize the release of drugs from microspheres through the change of fluorescence intensity ratio. Furthermore, this FRET method can be potentially used to visualize the drug release in vivo.

  1. Looking at the way that the Authors employed the FRET method, it seems to me that the FRET analysis provides more information on the disassociation of the PLGA sphere rather than the release kinetics of the drugs. Should Authors reconsider what is actually quantified here?

Author Response:Thank you for your insightful comment. Indeed, the FRET analysis provides more information on the disassociation of the PLGA sphere rather than the release kinetics of the drugs. However, the model drugs used in this study were hydrophobic ones. The release of the drug molecules from the microspheres were mainly attributed to the erosion and degradation of the PLGA matrix. Therefore, this erosion of the matrix bridged the FRET changes and drug release, making the quantification of the drug release by FRET changes possible.  

Reviewer 2 Report

Comments and Suggestions for Authors

The presented work is devoted to the development of a methodology for quantitative estimation of the kinetic patterns of drug release from polymer microspheres during their degradation in an aqueous environment. The method used is based on the FRET effect (dipole-dipole energy transfer from one fluorescent dye to another). This process is very sensitive to the distance between the donor and acceptor molecules. When the microsphere is destroyed, the donor and acceptor are released into the external environment, the distance between them increases and the fluorescence intensity decreases. When the microsphere is completely destroyed, fluorescence disappears.

The presented work was done very high quality and, one might say, meticulously. The microspheres were characterized by a number of methods. The obtained correlation between the fluorescence intensity and the drug release appears reliable.

However, in my opinion, the text of the article needs significant improvement.

1) The kinetic regularities of the release can differ significantly for drug molecules and for fluorescent dyes. Large molecules would be released from swollen microspheres slowly than smaller ones. This effect is partially manifested in the systems under study (see Figure 4A,B). It is necessary to emphasize this circumstance.

2) The reason for the increase in fluorescence intensity of both donor and FRET during the first five days is unclear (Fig. 3B,E)

3) Most of the text is a description of the figures in words. In my opinion, this is unnecessary. Readers are able to study the figures, but authors should focus on formulating the main conclusions.

4) The text of the article must be carefully edited and shown to a native English speaker. Now, it is difficult to read and understand (see, for example, lines 195-196). There is an error in the caption in Figure 6.

Comments on the Quality of English Language

English Language needs to be improved.

Author Response

  1. The kinetic regularities of the release can differ significantly for drug molecules and for fluorescent dyes. Large molecules would be released from swollen microspheres slowly than smaller ones. This effect is partially manifested in the systems under study (see Figure 4A,B). It is necessary to emphasize this circumstance.

Author Response:Yes, indeed the larger molecules would be releases more slowly compared to the smaller ones. In this study we still think that the molecular weights of risperidone and clozapine are similar, although the amine groups on clozapine can catalyze the degradation of the PLGA to a greater extent. This phenomenon was explained in Figure 4C, the text of which is embodied in 331-342 on page 9.

  1. The reason for the increase in fluorescence intensity of both donor and FRET during the first five days is unclear (Fig. 3B,E)

Author Response: We explain this phenomenon at 298-302 on page 8. We believe that this phenomenon is due to the fact that the concentrations of fluorescent dyes were relatively high within the microspheres at the first a few days, which causes self-quenching, followed by a decrease in fluorescence concentration within the microspheres due to the continuous release of the drug and fluorescent dyes from the microspheres, and the quenching effect is diminished. Although this effect was greatly alleviated by the use of calibrated FRET/D value to represent FRET changes, we will optimize the ideal concentrations of the fluorescent dyes to improve this in the future study.

  1. Most of the text is a description of the figures in words. In my opinion, this is unnecessary. Readers are able to study the figures, but authors should focus on formulating the main conclusions.

Author Response:Thank you for your suggestion. We have streamlined the text and added more content of discussions and conclusions. Such as lines 258-260 on page 6, lines 293-294, 298-307 on page 8, lines 356-358 on page 10, and lines 374-375, 377-380 on page 11.

  1. The text of the article must be carefully edited and shown to a native English speaker. Now, it is difficult to read and understand (see, for example, lines 195-196). There is an error in the caption in Figure 6.

Author Response:Thank you for your suggestion. We have thoroughly polished the language for the whole manuscript. Those changes included but not limited to formatting error on lines 196-198, Page 2, line 65, line 244 on page 6, line 271 on page 7 has been corrected, the error in Figure 6 has been corrected. On page 1, line 41, the redundant symbols have been removed, etc.

Reviewer 3 Report

Comments and Suggestions for Authors

The MS “Quantification of microsphere drug release by fluorescence imaging with the FRET system” Is well done job of being able to visualize drug release from the vehicle. The large amount of work done and the thoughtfulness of the results are visible. The literature is well selected and matches the theme of the manuscript.

There are a few points to clarify

- In the abstract you use drug abbreviations. Please enter them before use.

- Line 122 extra dot before link

- In the experimental part, it is not entirely clear in what quantity, when and in what ratio dyes are added to each other and drugs

- Line 153 NaN  is NaN3 - Sodium Azide I supposed

- There are doubts about sodium azide and Tween. Do they affect the fluorescence of dyes? It's worth checking out. Both are preservatives and provide a fairly aggressive environment that can affect both dye fluorescence and drug release studies

- The abbreviation in Figure 2D is unclear

- Figure 3B. To confirm this theory, pure preparations must be analyzed at a given concentration. Receiving 3C is unclear and does not correlate with 3D

- Figure 7: Is the presence of FRET for 30 days an indicator that some of the capsules have not disintegrated?

- The classic way to show the effectiveness of quenching on concentration is the Stern-Volmer method. Is it possible to use it here or is the presence of changes in donor and acceptor concentrations a limiting factor?

Author Response

  1. In the abstract you use drug abbreviations. Please enter them before use.

Author Response:Thank you for your suggestion. Abbreviations have been added to lines 25-26 on the first page.

  1. Line 122 extra dot before link

Author Response:Thank you for your suggestion, it has been corrected.

  1. In the experimental part, it is not entirely clear in what quantity, when and in what ratio dyes are added to each other and drugs

Author Response:Thank you for your suggestion, more information was added on page 3, line 111-114.

  1. Line 153 NaN is NaN3 - Sodium Azide I supposed

Author Response:Thank you for pointing that out, it has been corrected on page 4, line 157.

  1. There are doubts about sodium azide and Tween. Do they affect the fluorescence of dyes? It's worth checking out. Both are preservatives and provide a fairly aggressive environment that can affect both dye fluorescence and drug release studies

Author Response:Thank you for the nice question. From the report of literatures, moderate concentrations of tween and sodium azide were commonly used in the release medium of the in vitro release experiment of microspheres to simulate the in vivo physiological environment. We did tested the existence of these two substances did not influenced the drug release nor the intensities of FRET dyes. The results of the fluorescent or confocal experiment also proved the robust reading of fluorescence during the release experiments.

  1. The abbreviation in Figure 2D is unclear.

Author Response:Thanks for the suggestion, the explanation of the chart is on page 8, lines 287-289.

  1. Figure 3B. To confirm this theory, pure preparations must be analyzed at a given concentration. Receiving 3C is unclear and does not correlate with 3D

Author Response:Thanks to your comment. Yes the preparations analyzed in the study were at the same concentration. Figure 3D to 3F actually showed the results for clozapine samples, while 3A to 3C showed for risperidone samples.

  1. Figure 7: Is the presence of FRET for 30 days an indicator that some of the capsules have not disintegrated?

Author Response:Yes, we do think there are still traces amount of microspheres are still incompletely degraded, although the drug release cannot be measured from HPLC. This is also a sign that fluorescence signal can be more sensitive to measure compared to measuring drug content from the release media.

  1. The classic way to show the effectiveness of quenching on concentration is the Stern-Volmer method. Is it possible to use it here or is the presence of changes in donor and acceptor concentrations a limiting factor?

Author Response:Thank you for the valuable comment. The Stern-Volmer method reflects the relationship between the quantum yield or fluorescence concentration and the concentration of the substance that quenches the fluorescence, i.e., Cy5. We assume it will be difficult to quantitatively measure the quenching effect using this method under the setting of this study. This is a very inspiring thought and we may consider this method in our future study.